# A Review of Permeability and Flow Simulation for Liquid Composite Moulding of Plant Fibre Composites

**DOI:** 10.3390/ma13214811

**Published:** 2020-10-28

**Authors:** Delphin Pantaloni, Alain Bourmaud, Christophe Baley, Mike J. Clifford, Michael H. Ramage, Darshil U. Shah

**Affiliations:** 1Research Institute Dupuy De Lôme (IRDL), Université Bretagne Sud, UMR CNRS 6027 Lorient, France; delphin.pantaloni@univ-ubs.fr (D.P.); alain.bourmaud@univ-ubs.fr (A.B.); christophe.baley@univ-ubs.fr (C.B.); 2Department of M3, Faculty of Engineering, University of Nottingham, Nottingham NG7 2RD, UK; Mike.Clifford@nottingham.ac.uk; 3Centre for Natural Material Innovation, Department of Architecture, University of Cambridge, Cambridge CB2 1PX, UK; mhr29@cam.ac.uk

**Keywords:** polymer matrix composites (PMCs), liquid composite moulding (LCM), resin transfer moulding (RTM), permeability, flow modelling, natural fibres, biocomposites

## Abstract

Liquid composite moulding (LCM) of plant fibre composites has gained much attention for the development of structural biobased composites. To produce quality composites, better understanding of the resin impregnation process and flow behaviour in plant fibre reinforcements is vital. By reviewing the literature, we aim to identify key plant fibre reinforcement-specific factors that influence, if not govern, the mould filling stage during LCM of plant fibre composites. In particular, the differences in structure (physical and biochemical) for plant and synthetic fibres, their semi-products (i.e., yarns and rovings), and their mats and textiles are shown to have a perceptible effect on their compaction, in-plane permeability, and processing via LCM. In addition to examining the effects of dual-scale flow, resin absorption, (subsequent) fibre swelling, capillarity, and time-dependent saturated and unsaturated permeability that are specific to plant fibre reinforcements, we also review the various models utilised to predict and simulate resin impregnation during LCM of plant fibre composites.

## 1. Introduction

### 1.1. Liquid Composite Moulding (LCM)

Resin infusion or liquid composite moulding (LCM) processes accounted for 11% (ca. 1.3 Mtonnes) of the 2019 global composite production market [1]. While the ratio of thermoset-to-thermoplastic composites has evolved from 98:2 in the 1980s to almost 60:40 at present (2019), the use of a wide range of principally thermosetting resin-based LCM processes [2], such as vacuum infusion and resin transfer moulding, has grown consistently over the last few years and decades. The latter is primarily due to a decline in manual processes such as hand lay-up and spray-up [1].

The basic approach in any LCM process is to force a catalysed thermosetting liquid resin, such as epoxy, polyester, vinylester, phenolic, or furan resin, to flow through a dry, stationary, porous, compacted reinforcement inside a closed mould by creating a pressure differential between the inlet(s) and outlet(s). In general, a LCM process can be divided into four stages: (i) reinforcement lay-up, (ii) mould filling, (iii) post-filling, and (iv) demoulding. As the primary aim of any LCM process is to ensure complete filling of the mould, successful execution of LCM involves understanding, controlling, and optimising the mould filling stage in particular. This stage dictates the production cycle time (for small to medium sized parts), quality (viz. defects including voids and dry spots), geometry (e.g., thickness), and ultimately mechanical properties of the final part. Not surprisingly, computational mould filling simulations are widely used as a cost-efficient and time-saving tool to optimise the LCM process [3,4]. However, accurate manufacturing process simulations require accurate input data. Here, we consider the processing of plant-fibre-reinforced plastics via LCM.

### 1.2. Plant Fibre Composites in LCM Processes

Plant fibre composites accounted for over 11% of the 11.7 million tonnes global fibre-reinforced composite market in 2019 [1]. This includes plant fibres such as cotton, flax, hemp, jute, and kenaf, but excludes 6.1 Mtonnes of wood–plastic composites produced in the same year [1]. Indeed, the production of non-wood plant-fibre-reinforced plastics dwarfs the production of non-glass synthetic fibre (e.g., carbon, aramid) composites in terms of volume.

The increasing consideration of plant fibres as sustainable next-generation reinforcements requires tackling the first hurdle, which is composite manufacture (reviewed in [5,6,7,8,9,10]). Due to the commercial applications of plant fibre composites in predominantly small-sized, high-volume, low cycle time, non-structural components, such as for decking for the construction sector, as well as interior panels for the automotive sector, injection–extrusion moulding and compression moulding are the widely used manufacturing techniques [7,11]. The reinforcement forms have typically been pellets or granules for the former and random fibre mats for the latter. While wood- and cotton-reinforced composites largely (>80%) employ thermosetting matrices, plant fibre composites are primarily (>70%) based on thermoplastic matrices [7,11,12].

LCM, on the other hand, is particularly suitable for (semi-)structural components utilising textile reinforcements, which are fabrics comprising aligned, continuous yarns or tows that are knitted, woven, stitched, or braided in thermosetting matrices with high fibre content. Other than the potential to produce high-performance composites, LCM processes are specifically well-suited to natural fibre reinforcements for a variety of reasons [7,13,14]:Low processing temperatures (often ambient, if not <150 °C), thereby avoiding thermal degradation of plant fibres (which decompose above ca 200 °C, although both temperature and time of exposure are pertinent parameters [15]);Minimal fibre damage during composite processing due to low shear rate range (as opposed to injection–extrusion moulding), thereby allowing retention of high reinforcement length, alignment, and mechanical properties;Use of liquid resins with low viscosities (0.1–1 Pa·s), thereby allowing good preform impregnation with low porosity, even at low compaction or injection pressures;Relatively low-cost tooling, making the process compatible with low-cost plant fibres, particularly when manufacturing in low- and middle-income countries with an abundance of indigenous plant fibres;Closed-mould LCM processes reduce exposure to harmful emissions, therefore offering worker-friendly conditions, much like non-hazardous plant fibres.

Consequently, LCM of plant fibre composites has received much recent attention in scientific research, where critical aspects such as reinforcement compaction [13,16,17,18,19,20,21], permeability [17,19,20,22,23,24], wetting, and resin flow behaviour [19,22,24,25,26] have been investigated.

Here, we aim to critically review the literature on LCM of plant fibre composites. It is expected that the differences in structure, morphology, and composition of plant and synthetic fibres, their semi-products (i.e., yarns and rovings), and their textiles will have a perceptible effect on their processing via LCM. We aim to identify key findings concerning the reinforcement-related factors that influence the mould filling stage during LCM of plant fibre composites. We specifically review the permeability of plant fibre reinforcements and highlight their key differences from synthetic fibre reinforcements. These insights are then used as foundations to discuss, in a simplified manner, developments in the numerical and computational modelling of the mould filling stage in LCM of plant fibre reinforcements using adapted resin flow models. An overarching aim of our review is to support better control, optimisation, and simulation of plant fibre composites manufactured by LCM processes, and thereby progress the development of plant fibre composite products in wider applications with reduced waste and efficient processing. The review has been written for an informed composites audience, and will be particularly of interest to the biocomposites communities in academia and industry.

## 2. Plant Fibre Reinforcements in LCM Processes

The key reinforcement-related parameters that affect the complex mould filling process are preform compaction and preform permeability. Extensive research has been conducted to measure, predict, and simulate the mould filling process in the LCM of conventional synthetic fibre composites [3]. Plant fibre reinforcements, however, require specific considerations.

### 2.1. Specificities of Plant Fibre Reinforcements

While synthetic fibres generally have very regular morphologies and no internal porosities, this is not the case for plant fibres; they exhibit a range of specificities potentially influencing their impregnation by a liquid resin. Notably, each plant fibre is an individual cell, and these cells may have specific and distinct functions based on their role in the plant. Some plant fibres, such as flax, have purely a structural support role, whereas others, such as wood tracheids, can have mechanical support and conduction roles, for example. Two main aspects can be considered: their morphology and their surface properties.

#### 2.1.1. Plant Fibre Morphology

Plant fibres are elementary cells made up of successive layers that differ in composition and ultrastructure [27]. Typically, a plant fibre consists of a primary wall on the outside and a secondary wall divided into three sub-layers (S1, S2-G, and S3-Gn). The S2-G layer is generally the thickest. The main property-governing parameters of each layer are their crystalline cellulose content and the cellulose microfibrillar angle (MFA); these can differ significantly from layer to layer. These two endogen parameters determine fibre performance, and in particular tensile longitudinal mechanical properties.

However, these structural parameters have little impact on the resin impregnation behaviour during LCM. Rather, the architecture of the fibre, and in particular its central cavity, called the lumen, has a greater role. The surface roughness of the fibre may be affected by the fibre extraction methods, and particularly the quality of the retting process (Figure 1a,b). Indeed, depending on the type of fibre, the lumen can have a significant volume, presenting a resin migration pathway during the impregnation stage [28]. The size of the lumen may represent only a small part of the cross-sectional area, as in the case of flax (Figure 1c) and hemp, or in contrast it may represent a significant part for kenaf and wood (Figure 1d,e) [27].

This phenomenon will be more pronounced for shorter cut fibres, as lumen access areas will be more numerous (per unit length). As with the size of the lumens, the length of the elementary fibre also varies significantly with plant species. This is principally linked to the origin of the fibre tissue, as well as the mode of growth (ontogenesis). Among the longest fibres, the primary fibres of flax, hemp, and nettle are formed through intrusive growth, which enables them to reach lengths of several tens of mm. These fibres are elementary cells, but they have the particularity of multiplying their nucleus, which enables them to reach these large dimensions [29]. For other species, and in particular when the fibre cells play a protective role or are responsible for fluid transport, their dimensions are generally limited to a few mm; this is the case for wood, jute, and sisal fibres.

The cross-sectional morphology of the fibres can also have an important influence on their impregnation behaviour, particularly when fibre preforms (e.g., mats, textiles) are compacted to achieve higher fibre volume fractions. Synthetic fibres are generally circular, which leads to specific geometric packing limits in terms of fibre volume fractions, e.g., π/4 = 0.785 for a square packing arrangement. As far as plant-derived fibres are concerned, their cross-section is generally polygonal, as evidenced for flax in Figure 1c. Such cross-sections make it possible to achieve high fibre volume fractions within plant structures (see Figure 1c), such as stems. Plant structures can, therefore, be a source of bioinspiration to optimise the fibre content in composite materials [30]. Indeed, natural plant structures have the advantage of being “grown’” through intelligent self-assembly processes. However, in a conventionally fabricated fibre composite, it is extremely difficult to reproduce these near-perfect packing arrangements due to the irregular cross-section and discrete length of these plant fibres, as well as the crude nature of the manufacturing processes.

#### 2.1.2. Fibre Surface Properties

In addition to morphology, the surface quality (vis. roughness) and composition (relating to surface energy and wettability) of the outer cell wall layers plays an important role in the flow and impregnation phenomena. Plant fibres present notable differences in biochemical composition across two large families: (i) gelatinous fibres (flax, hemp, and nettle, for example), which are mainly composed of crystalline cellulose and a reduced fraction of non-cellulosic polysaccharides; and (ii) xylane-type fibres (jute, normal wood, and kenaf, for example), which are made up of fairly equal fractions of cellulose, lignin, and non-cellulosic polysaccharides [29]. While the overall composition of the cell walls has little impact on the impregnation properties or on the diffusion of the resin, the composition and surface of the external layer directly in contact with the resin do affect flow.

After mechanical extraction, the surface quality of the plant fibres and their bundles generally depends on the degree of retting, which in turn is highly dependent on weather conditions and farmer know-how. Indeed, in the case of imperfect retting, residues of middle lamellae (Figure 1b) are present on the surface of the extracted fibres. This can not only be detrimental to the quality of the interface with the resin [31], but can also cause disturbance during LCM by causing collapse of the channels or flow paths of the resin [32]. Indeed, these intermediate lamellae possess different stiffnesses and degrees of cohesion [33], which can inhibit the flows.

Significant differences in composition have been highlighted between the surfaces of flax and hemp fibres [34], with hemp fibres having higher lignin levels, for example. It should be noted that this lignification can occur in flax if the plants are harvested late. Fibres such as jute and kenaf also have high levels of lignification. It has been shown that this difference in composition allows hemp to have better wettability with polyolefins [34]. Hence, the nature of the outer wall, depending on its lignin content, will influence the flow, depending on the chemical nature of the resins used. In general, plant fibres display numerous polar hydroxyl groups on the surface, which have a strong affinity with aqueous media. This point should be carefully considered to avoid fibre swelling and resin absorption phenomena during the composite material manufacturing process; this will be discussed further in later sections.

### 2.2. Compaction and Packing of Plant Fibre Reinforcements

In relation to composite manufacture, the compaction behaviour of technical textiles affects the reinforcement permeability and part fill time in the mould filling process, and also determines the thickness and volumetric composition (i.e., fibre volume fraction) of the final part [35]. Tight control of the part thickness (and therefore weight) is a requisite for quality assurance in any composite manufacturing process. In addition, in their uncompressed state, textile reinforcements have a low fibre volume fraction (typically between 10 and 25% [35]); for semi-structural applications this must be increased (to up to 70%) during processing to exploit the mechanical properties of the reinforcement. Studying the relationship between compaction pressure *P* and fibre volume fraction *v_f_* for a given preform also enables determination of the maximum (practical) fibre volume fraction. Consequently, compaction plays an important role in LCM processes and in the stamping of textile-reinforced thermoplastic composites. While the compaction characteristics of synthetic reinforcements is well-studied [35], the compaction response of natural fibre reinforcements is a relatively new topic [13,16,17,18,19,20,21].

Analytical studies have provided an understanding of the key mechanisms driving compaction [36,37,38]. Table 1 identifies the primary and secondary mechanisms contributing to the compaction at three key scales—the fibre scale, yarn scale, and preform scale.

As shown in Figure 2, plant fibre preforms experience all the compaction mechanisms that synthetic fibre preforms experience (described in Table 1), but not vice versa. In particular, cross-section deformation at the fibre scale (Table 1) is limited to plant fibre reinforcements, which due to their hollow nature and low fibre transverse stiffness and strength [7,39] undergo lumen closure and transverse cell wall buckling and delamination during compaction [13,17,40]. This is illustrated in Figure 3. A substantial change in fibre cross-section shape due to the low fibre transverse stiffness—a result of the hollow nature of the fibre and anisotropic properties of cellulose—is important in the context of preform compaction, as it may alter the potential for fibre relative motion and yarn reorganisation, and could lead to hindered impregnation in localised inter-fibre zones.

Lundquist et al. [40] found that lumen compression occurred at fibre volume fractions of 34% to 69% in wood pulp random mat preforms (Figure 3). Void condensation is a dominant compaction mechanism, particularly at low compaction pressures (e.g., when *v_f_* < 55%) in such random mat preforms, with fibre or yarn bending deformation and flattening being additional secondary mechanisms, particularly at high compaction pressures (e.g., when *v_f_* > 55%). Francucci et al. [13] have also observed such irreversible transverse cell wall deformation in compacted woven jute fabrics, and this phenomenon increased as the fibre content increased (Figure 3). They noted that this mechanism would contribute, alongside irreversible yarn cross-section deformation, yarn flattening, and yarn nesting, to the compaction of the woven material. However, Francucci et al. [13] opined that such lumen collapse would mostly occur when fibre rearrangement and tow movements are limited, i.e., at high fibre volume fractions. This is in stark contrast with the observation of Lundquist et al. [40] (Figure 3), where all wood pulp lumen had collapsed by *v_f_* = 69%, and the wood pulp random mat was compressed further up to *v_f_* = 90%.

Luminal porosity varies between plant species and has a significant effect on the transverse stiffness and yield strength in both compression and tension [40,44,45], where fibres with a larger lumen (and smaller second moment of area) tend to deform more readily. Naturally, therefore, different plant fibres would exhibit different degrees of fibre bending and cross-section deformation during plant fibre preform compaction. Other micro- and macro-structural features, such as fibre cross-section shape and surface roughness (relating to fibre slippage) and degree of fibre alignment and dispersion, affect which (and when) compaction mechanisms will play primary and secondary roles.

### 2.3. In-Plane Permeability of Plant Fibre Reinforcements

Permeability is defined as the ease of fluid flow through a preform, and therefore it is an inverse measure of the flow resistance [3]. The greater the preform permeability, the easier it is for the resin to impregnate the reinforcement, and the lesser the time needed to fill the mould. While permeability characterisation of natural fibre reinforcements is a relatively new topic [17,19,20,22,23,24], researchers have already found some critical differences between natural and synthetic fibre reinforcements. In this review, we focus on in-plane permeability as opposed to through-thickness permeability, as almost all research into plant fibre permeability is on this area.

The limited research on permeability studies of natural fibre reinforcements shows that while plant fibre reinforcement exhibit higher permeability than glass fibre reinforcements (Figure 4) [17,23], wood fibre mats exhibit significantly lower (by about two orders of magnitude) permeability than glass fibre mats [46]. Table 2 lists permeability data of various plant fibre reinforcements studied in literature, by showing permeability K values at specific fibre volume fractions (*v_f_* = 0.2 and 0.5). Table 2 also lists semi-empirically derived Carman–Kozeny constants, C and n, which are commonly used to describe permeability–porosity relationship in synthetic and plant fibre composites.

A predominant number of studies have used mineral oil (non-polar-fluid) or non-reactive glycerin solution (polar fluid) at fixed (often ambient) temperatures and viscosities (comparable to commercial resins) to study the permeability of plant fibre reinforcements. While the use of oils enable the study of permeability with no fluid absorption and consequent swelling of the plant fibres, the use of glycerin enables the study of “sink” and “source” effects (described in more detail in Section 3.2). A few studies have examined permeability or flow behaviour with commercial thermosetting resins such as phenolics [25] at elevated temperatures (e.g., 60 °C). However, even in these cases viscosity is (assumed to be) constant during the mould filling stage, as the fill time (of 60 s to 1200 s) is significantly shorter than the extended pot life (several hours) enabled by the selected catalysts.

Researchers have hypothesised that the higher flexibility of plant fibres in comparison to glass fibres may make the former more permeable [23]. On the other hand, the lower permeability of wood fibre mats (in comparison to glass fibre mats) has been attributed to the more tortuous flow paths in the less-efficiently packed wood fibre mats, owing to the very short length of wood fibres [22]. The argument may have merit, as flax yarn random mats composed of longer fibres have notably and consistently higher (saturated) permeability across a range of fibre volume fractions [20]. For instance, the permeability values of random mats with 50 mm fibre lengths were found to be 22% and 25% higher at the lowest (*v_f_* = 0.2) and highest (*v_f_* = 0.4) fibre volume fractions, in comparison to random mats with 15 mm fibre lengths [20]. Shives, which are present in flax mats, may also increase the permeability due to their porous structure by creating channels for fluid movement, blurring the influence of the volume fraction [52].

Yarn diameter (or linear density) is also known to affect the permeability of flax yarn random mats [20]. Umer et al. [20] found that the permeability of medium yarn diameter (0.56 mm) mats was consistently (i.e., over a range of porosity levels) 27% higher than small yarn diameter (0.35 mm) mats, however the permeability of small and medium yarn diameter mats was consistently 68–77% higher than large yarn diameter (0.81 mm) mats. Given that the permeability of preforms is dominated by the characteristics of open channels, and that preform geometric parameters that describe the characteristics of the open channels include the number of fibres in the bundle, the twist angle and orientation of the bundle, and the dimensions and cross-sectional shapes of bundles and single fibres, Umer et al. suggested that large diameter yarns were less compact and had lower twist levels. The loose fibres on the surface of large diameter yarns restrict fluid flow through the open channels, thereby decreasing permeability. The twist level of yarns may also affect permeability and impregnability by altering competition between micro-flow and macro-flow [20,41,53]. This dual-scale flow is reviewed further in a later section.

It is evident from Table 2 that experimental conditions have a notable effect on the permeability data obtained for even the same plant fibre reinforcement (e.g., jute plain-woven fabric). In particular, it is important to clarify (i) the type of test fluid employed (and its polarity) and ii) the viscosity of the test fluid, and (iii) whether the saturated or unsaturated permeability has been measured. For plant fibre preforms, saturated permeability tends to be higher than unsaturated permeability [24], and permeability tends to be lower when measured in less viscous fluids (which is expected and easily modelled). The more interesting issue is of the polarity of the test fluid, as plant fibres, unlike synthetic fibres, absorb polar fluids, and consequently swell and soften [24,54].

Plant fibres are different from synthetic fibres such as glass, in that natural fibres tend to be polar and hydrophilic; that is, plant fibres absorb polar liquids. This not only includes liquids used during compaction and permeability testing such as water and glycerin solution, but also includes polar thermosetting resins such as vinylester and phenolics [24]. Moreover, plant fibres swell upon liquid absorption (10–25% moisture regain) [24]. For instance, over an immersion time of 6000 seconds, jute fibres exhibited equilibrium transverse swelling (i.e., change in diameter) by over 18% in glycerin solution, and between 6 to 8% in vinylester and phenolic resin [24]. In comparison, glass fibres do not absorb liquid (<2% moisture regain), nor do they exhibit swelling [24].

Conceivably, fluid absorption and swelling are important mechanisms in plant fibre preforms, due to which both saturated and unsaturated permeability are reduced. Francucci et al. [24] demonstrated through absorption, swelling, and permeability tests on untreated and treated jute fibre random mats that untreated mats had up to five times higher absorption levels, up to twenty times higher transverse swelling, and an order of magnitude lower permeability than treated mats. Umer et al. [54] show that permeability levels of wood fibre mats measured in (polar) glucose syrup were lower than that measured in (non-polar) mineral oil. Fluid absorption removes fluid from the main stream, acting as a sink component, and thus decreasing flow velocity during the unsaturated flow. The swelling of the plant fibres constricts the open flow path, reduces porosity, and increases flow resistance during saturated flow. Since plant fibres will behave differently with different liquids, to accurately measure their permeability, the plant fibre reinforcement should ideally be studied with the particular resin that is to be used during LCM as the test fluid.

#### 2.3.1. Permeability Anisotropy: Direction Matters

Aligned reinforcements, such as non-woven and woven fabrics, tend to produce anisotropic flow due to anisotropic permeability. The permeability anisotropy ratio of plain-to-woven jute fabric has been estimated to be 1.24 [17]. Random mats are quasi-isotropic reinforcements and would, therefore, produce quasi-isotropic flow due to quasi-isotropic permeability. Xue et al. [19] measured the anisotropy ratio in permeability (Kmax/Kmin) to range between 1.08 and 1.46 for flax random mats; cross-laid mats were more isotropic than parallel-laid mats. Mekic et al. obtained anisotropy ratios ranging between 1.58 and 1.71 for a range of in-plane isotropic flax fibre non-woven preforms [55]. Orientation distribution analysis showed that fibres were uniformly distributed in cross-laid mats, but fibres in parallel-laid mats were oriented primarily along the machine direction [19]. As the preferential resin flow path (path of least resistance) would be along the principal fibre orientation direction, permeability was much higher in this direction for parallel-laid mats. In cross-laid mats, due to the fairly equal paths of least resistance, flow was more isotropic. Note that interestingly, parallel-laid mats exhibited lower overall permeability than cross-laid mats compacted under the same pressure, due to fibre nesting and consequently lower porosity in parallel-laid mats. Looking at woven fabrics, permeability is anisotropic due to the fibre orientation. In modifying this orientation by shearing fabrics, which is representative of an industrial warping process, the permeability response, the anisotropy behaviour, and the preferential impregnation direction of the fabric are impacted [51].

### 2.4. Summary

This section reviewed the specificities of plant fibre reinforcements that affect their compaction, permeability, and impregnation through LCM methods. Plant fibres have unique morphologies, characterised by the presence of luminal and structural porosity (which are significant in specific plant fibre types) and their discrete lengths. These factors lead to compaction mechanisms unique to plant fibre reinforcements, such as luminal collapse and transverse cell wall deformation, alongside irreversible yarn cross-section deformation and flattening, as well as nesting and packing of the reinforcement fabric, at high compaction pressures and fibre volume fractions. The morphologies of plant fibres and their compaction behaviour influence their permeability. Reinforcements with longer fibres, moderate yarn diameters, aligned textiles (versus random non-woven mats), and higher permeability (lower fibre volume fractions) have higher permeability, increasing further with the viscosity of the resin and with prior saturation of the reinforcement. Plant fibre reinforcements have substantially higher permeability than synthetic fibre reinforcements.

Various studies have assessed the in-plane permeability of plant fibre reinforcements, such as woven fabrics and non-woven mats, although for better control of the process (e.g., maintaining constant viscosity and temperature), studies primarily use mineral oil (non-polar) or glycerin solution (polar) as the test fluid, with viscosities comparable to thermosetting resins. Very few studies have examined the permeability of plant fibre reinforcements with catalysed thermosetting resins, particularly at elevated temperatures, and those that have done this have utilised fill times substantially shorter than the resin pot life. Plant fibres generally have rough fibre surfaces rich in functional groups such as hydroxyls, having an affinity with conventional LCM resins, such as epoxies, polyesters, vinylesters, and phenolics. There is notable evidence that plant fibres not only absorb water and glycerine and subsequently swell, but they react similarly to resins such as vinyl esters and epoxies. Quantification of the scale of resin absorption and swelling effects during impregnation is vital to accurate prediction, modelling, and simulation of the LCM flow behaviour of plant fibre reinforcements. The next section focuses on this.

## 3. Flow Modelling and Simulation of Natural Fibre Composites

Controlled and complete filling of the mould with adequate wetting of fibres is a primary objective in LCM. Poor fibre wetting would lead to poor mechanical properties due to micro-void formation and poor interfacial adhesion, while uncontrolled and incomplete filling would lead to defect formation (e.g., dry spots), poor part quality, and even part scrappage and material wastage. Optimising mould fill time whilst avoiding fluid pressure build-up is important in controlling mould filling. The microscopic and macroscopic flow of the liquid resin through gaps within yarns and tows and the porous preform, respectively, is, therefore, important to study. A number of factors affect the complex mould filling process, including the permeability, pressure differential in the mould cavity, rate of resin injection, resin viscosity (as a function of time), and inlet and outlet gate locations. Often permeability studies and Darcy’s law, which can be derived from the Navier–Stokes equation through averaging methods [56], are used to model the complicated viscous flow of resin in a porous media. Here, we explore the developments in the numerical and computational modelling of the mould filling stage in LCM of plant fibre reinforcements using adapted resin flow models.

### 3.1. Classical Flow Models

Darcy’s law generally describes the flow of Newtonian fluids in porous media. Assuming steady-state 1D flow, Darcy’s law (Equation (1)) states that the macroscopic volumetric flow rate *Q* is proportional to the mould cross-section area *A* and the pressure difference over the sample Δ*P*, and inversely proportional to the sample length *L* (in the flow direction) and fluid viscosity *μ*. The constant *K* is termed the permeability.
(1)$$Q=K\frac{A}{L}\frac{\Delta P}{\mu }$$

Most flow behaviour studies on plant fibre composites use Darcy’s law to semi-empirically determine reinforcement permeability, *K*. Permeability can be experimentally measured by tracking the progression of the fluid flow front and monitoring the pressure field (Figure 5). A planar flow cell is used, in which a test fluid (ideally with a similar viscosity to the resin) is injected into the fabric. A 1D unidirectional flow achieved through line gate injection (Figure 5a,b) is commonly used. However, a 2D radial flow achieved through central injection (Figure 5c,d) may be more appropriate for anisotropic reinforcements to quickly measure anisotropy in permeability. The fluid viscosity is measured using a viscosimeter or rheometer. A flow meter, which is typically placed at the inlet or outlet port, may be used to measure fluid velocity. Alternatively, the flow velocity may be estimated by monitoring the flow front evolution. The flow front can be tracked manually (e.g., with scale bars, as in Figure 5b), with a video camera, or through pressure sensors. The pressure gradient in the mould may be recorded through manometers or pressure transducers, which are typically placed at the inlet and outlet ports. The pressure difference is measured relative to the flow front (which is at atmospheric pressure).

The saturated (or steady-state) permeability, Ksat, is obtained once the reinforcement is fully saturated using Equation (2). On the other hand, the unsaturated (or transient) permeability, Kunsat, is obtained when the flow profile is measured during the injection process. Darcy’s law in Equation (1) can be integrated and rearranged to the form in Equation (3) [58], where *x* is the position of the flow front at time *t*, *ϕ* is the fabric porosity, and *v* is the average fluid velocity. Note that fabric porosity is directly related to the fibre volume fraction. For a demonstration, Figure 6a shows the linear relationship between the square of the flow front position and the fill time for two separate infusion conditions; the slope of the curve *x*^2^/*t* is used as an input in Equation (3) to determine Kunsat.
(2)$$K_{sat}=Q\frac{\mu }{\Delta P}\frac{L}{A}$$
(3)$$K_{unsat}=\frac{x^{2}}{t}\frac{\phi \mu }{2\Delta P},\;\mathrm{where}\;\phi =\frac{Q}{vA}\;\mathrm{and}\;\phi \equiv 1-v_{f}$$

The saturated and unsaturated permeability can then be determined over various conditions, for example varying fabric porosities (or fibre volume fraction), as shown in Figure 6b. In fact, the reinforcement fibre volume fraction, alongside the reinforcement type and orientation, is a key reinforcement-related factor affecting permeability. Increasing the fibre content (decreasing porosity) reduces the number of flow paths, and therefore reduces permeability. One way to play with this porosity is to compact the preform by varying degrees. Often, the three-parameter exponential function (Equation (4); proposed by Gauvin et al. [59]) or the two-parameter modified Carman–Kozeny equation [60] (Equation (5)) are used to model the permeability–porosity relationship. Both models are found to be well-suited to natural fibre reinforcements [17,23]. As seen in Table 2 for a range of plant fibre preforms, *n* in Equation (5) is close to 2 in most cases, implying that the fluid flow behaviour in natural fibre reinforcements is close to the original Carman–Kozeny model, where fibre arrangement can be described as parallel tubes with low tortuosity [23].
(4)$$K=a+b\cdot \exp^{c\phi }$$
(5)$$K=\frac{\phi^{n+1}}{C{\left(1-\phi \right)}^{n}}=\frac{{\left(1-v_{f}\right)}^{n+1}}{Cv_{f}{}^{n}}$$

Two problems commonly associated with permeability testing are mould deflection (induced by the pressure gradient on thin moulds) and uncontrolled flow [57]. The latter may manifest in the form of fibre washing or edge effects. Fibre washing refers to the displacement of the preform during filling, while edge effects, also known as race tracking, refer to the faster flow of resin at the edges due to lower permeability resulting from a clearance between the fibre preform and the mould edge. Indeed, such concerns have been voiced in studies on the mould filling process of plant fibre reinforcements as well (Figure 7) [6,23,24,25]. Fibre washing is more prevalent at high injection pressures and low fibre volume fractions (e.g., fewer reinforcement layers), while edge effects are more prevalent at high fibre volume fractions [25].

#### 3.1.1. Dual-Scale Flow and Capillarity

As a preform is commonly made of yarns, its impregnation usually involves a dual-scale flow. Resin flow between yarns (inter-yarn) is referred to as macro-flow, while resin flow through the yarns (intra-yarn) is referred to as micro-flow. As resin flows at low Reynolds numbers, inertial forces can be neglected. Macro-flow is dominated by viscous flow of the resin, while micro-flow is driven by capillary pressure developed within the tows [61]. Capillary effects are important to study, not least because they play a key role in the mechanism of void formation in LCM [61].

As Figure 8 highlights, at low flow velocities (and high fibre volume fractions) capillary flow dominates, leading to inter-yarn voids, while at high flow velocities (and low fibre volume fractions) viscous flow dominates, leading to intra-yarn voids. Based on the fact that impregnation in composites often leads to non-wetting dynamic angles [62], the dual-scale flow is explained as follows. As the inter-yarn flow front advances, the resin impregnates transversally in the yarns. As the resin fills the yarn, the inter-yarn flow rate is reduced. If the inter-yarn flow rate is faster than the intra-yarn flow rate, the resin flow front is unable to completely saturate and fill the yarns, leading to the appearance of intra-yarn voids. This is typically the case of *K_sat_*/*K_unsat_* > 1. On the other hand, in the case (thanks to capillary effects) that intra-yarn flow rate is faster than the inter-yarn flow rate, yarns will saturate with resin, while the inter-yarn regions may remain unfilled. This may lead to notable porosity in the inter-yarn regions. This is the case of *K_unsat_*/*K_sat_* > 1. Both effects have been shown for plant fibre composites (Figure 8) as a function of the fibre volume fraction [41,63]. There exists an ideal flow velocity (viz. capillary number) at which capillary effect is optimal, leading to equal flow rates in the inter- and intra-yarn regions, and therefore no (or minimal) void formation [61,62].

#### 3.1.2. Simulating Impregnation in Plant Fibre Preforms by Classical Approaches

Darcy’s law (Equation (6)) and the continuity equation (Equation (7)) are commonly and conveniently used to describe the flow of thermosetting resins in porous reinforcements.
(6)$$\bar{v}=-\frac{K}{\mu }\nabla P$$
(7)$$\nabla \cdot \bar{v}=0$$
where $\bar{v}$ is the volume-averaged fluid velocity and ∇P is the applied pressure gradient.

Most permeability studies on natural fibre reinforcements so far have been conducted on the presumption of a valid 1D Darcy’s law. The mould filling of natural fibre reinforcements has been successfully simulated using the conventional model by some researchers, including Kong et al. [64]. They performed a resin flow analysis on the upper and lower parts of a flax–vinylester agricultural chemical storage tank to predict fill times (and select a suitable injection pressure) and ensure complete impregnation. A flow simulation of the lower part is illustrated in Figure 9. The simulations were backed by experiments. It appears that the fill time for the upper part was well predicted, while the fill time for the lower part, which has a more complex geometry, exhibits a relative difference of 20%. This difference might come from the flax fibres being equated to synthetic fibres, ignoring the specific behaviour of plant fibres during impregnation. The next section explores the effects of these plant fibre specificities on the modelling of flow behaviour.

### 3.2. Modifying Flow Models to Accommodate Plant Fibre Specificities

#### 3.2.1. Sink Effects

It has been observed for natural fibre reinforcements that the saturated permeability is higher than the unsaturated permeability. The *K_sat_*/*K_unsat_* ratio ranges between 1.2 and 1.8 for plain woven jute fabrics [24]. As with Pillai and Advani [65], Francucci et al. [24] argue that the jute yarns act as a “sink”, leading to delayed impregnation of the tows through transverse micro-flow, which reduces the average macro-flow velocity, thereby reducing the macro-scale permeability. As previously mentioned, this is typical of dual-scale permeability with a non-wetting dynamic angle. Not only do plant fibres yarns draw resin from the main flow, which synthetic yarns also do to an extent, plant fibres also absorb the resin. The latter has a marked effect on the flow rate.

This absorption of fluid by the natural reinforcements provides an explanation for the high saturated permeability [24]. Again, as the jute fibres and yarns act as sink components (because fibres absorb liquid and yarns have micro-pores), the unsaturated permeability is reduced. Once infusion is complete (i.e., no more pores within tows need to be filled and fibres cannot absorb more fluid), the fibres and yarns are no longer sink components, and the permeability upon saturation is increased. Note that the swelling of the fibres upon saturation does restrict flow paths, and therefore the increase is only marginal. This argument is further strengthened by the fact that the difference between the saturated and unsaturated permeability vanishes with increasing porosity (decreasing fibre volume fraction) (Figure 6). In fact, at very high porosity (*ϕ* > 0.8), *K_sat_*/*K_unsat_* < 1 (Figure 6).

Notably, the difference between the saturated and unsaturated permeability is higher in untreated jute fabrics (*K_sat_*/*K_unsat_* = 1.70–1.84) than in treated jute fabrics (*K_sat_*/*K_unsat_* = 1.21–1.40) [24]. This suggests that while transverse micro-flow into tows may play an important role in the difference between the saturated and unsaturated permeability of plant fibre reinforcements, the polar nature of untreated natural fibres and their tendency to absorb polar fluids and swell has a more dominant role. Hence, the scale of the “sink effect” can be changed through targeted surface treatment of the plant fibres.

#### 3.2.2. Capillarity and Swelling Effects

Capillary effects in natural fibre reinforcements have received some attention [66,67], particularly due to the common notion that the hollow structure of plant fibres may enhance capillary effects. There is little evidence that resin can impregnate the lumen, especially if the fibres are not cut, and in most cases the lumen has been shown to remain unfilled after composite manufacture [16,41]. However, Yin et al. [68] systematically examined the effect of lumen impregnation on the flow front progress in a sisal composite. By comparing experimental measurements with model simulations, they demonstrated that the progress of the flow front in their experiments falls between the models considering full lumen impregnation and no lumen impregnation. One should note that sisal fibres have more substantial luminal porosity than bast fibres such as flax and hemp. The role of the lumen in the unsaturated permeability of a preform is probably dependant on the dimension of the lumen, which in turn is related to the nature of the fibre.

Investigations have revealed that the dynamic capillary pressure values in plain woven jute fabrics were −25 kPa and 36 kPa when measured in glycerin solution and vinylester resin, respectively [66]; that is, spontaneous infiltration occurs during infusion with glycerin solution, but capillary forces act against the flow during infusion with the resin. This has significant implications for experimental mould filling tests, as test fluids such as glycerin solution may provide deceivingly higher permeability and lower fill times.

The capillary pressure has been found to increase exponentially with fibre volume fraction [66]. Notably, jute fabrics consistently exhibit capillary pressures two to three times higher than synthetic reinforcements [66], implying that capillary effects and micro-flow are more dominant in natural fibre reinforcements. Francucci et al. [66] also demonstrated that the measured capillary pressure may be used to determine a corrected unsaturated permeability, which was found to be similar for both the test fluid and resin.

More recently, the impact of the absorption and the swelling on the capillarity behaviour of plant fibres has become a point of interest. It is well-known that plant fibres absorb moisture and water [69], but they have also been shown to absorb thermosetting resins. Testoni et al. [70] proposed a modified Washburn equation to predict the capillarity behaviour of flax bundles, taking into account the absorption of resin and the swelling as a function of time. Another model was developed by Vo [71], taking into account only the swelling, but considering the dual-scale impregnation of a composite. Both models are able to predict the capillarity behaviour of water with flax fibre, which is the severest fluid with regards to absorption and swelling. This absorption and swelling phenomenon can decrease the saturated permeability of a plant fibre fabric by 70%, depending on the resin used [49].

It appears that absorption (sink effects), capillarity, and fibre swelling are the main differences between synthetic and plant fibres. Absorption primarily influences the unsaturated permeability by increasing the sink effect, whereas swelling primarily impacts the saturated permeability by reducing the path for the resin.

#### 3.2.3. Models Suitable for Natural Fibre Reinforcements

During resin impregnation in an LCM process, plant fibre reinforcements, unlike synthetic reinforcements, absorb liquid and subsequently swell. As discussed previously, the percentage absorption and swelling is dependent on the (polarity of the) liquid; for instance, jute swells by 18–22% in glycerin solution but by 7–8% in vinylester or polyester resin [24]. Due to the absorption of liquid, plant fibre preforms effectively act as a “sink”, soaking up some of the resin that is infused. The subsequent fibre volume changes due to swelling have a two-fold, time-dependent, “source-like” effect: (i) they reduce the porosity, and thus the permeability; (ii) they increase the flow resistance. The conventional model in Equations (6) and (7) is, therefore, inappropriate, if not ineffective, in describing the permeability of, and resin flow behaviour in, plant fibre reinforcements, as it does not take these sink and source factors into account.

Several researchers have recently attempted to develop flow models specifically for plant fibre reinforcements, including [22,72,73,74,75,76]. A modified continuity equation (Equation (8)) was initially proposed, which incorporated a sink term *S*(*t*) related to the rate of fluid absorption by the reinforcement, and a porosity term *ϕ*’(*t*) related to the rate of decrease in preform porosity due to increase in fibre volume upon swelling, both of which are functions of time [22,72,75,76]. However, it was later found that sink and porosity terms had to cancel each other out to satisfy experimental observations [22,72]; that is, the rate of fluid absorption (per unit volume) had to be equal to the rate of the change of fibre volume (per unit volume). Consequently, Equation (8) is simplified to the conventional continuity equation of Equation (7).
(8)$$\nabla \cdot \bar{v}=-S(t)-\phi^{\prime }(t)$$

Following this, Masoodi et al. [18] proposed that the permeability *K*, alongside porosity *ϕ* and fibre diameter *d*, were only functions of time (Equation (9)), where the subscript 0 indicates initial values (at *t* = 0). Essentially, if the change in fibre diameter over time (due to swelling) was measured, fitted curves could be used to then predict the evolution of the porosity and permeability with time. Then, if the evolution of the pressure gradient as a function of time was known, Equation (10), which is derived by resolving Darcy’s law (Equation (6)) with the continuity equation (Equation (8)), could be used to predict the evolution of the flow front.
(9)$$K(t)=\frac{\phi {\left(t\right)}^{n+1}}{C{\left(1-\phi (t)\right)}^{n}},\text{​ }\mathrm{where}\;\phi (t)=1-(1-\phi_{0})\frac{d(t)}{d_{0}},\mathrm{and}\;d(t)=a\exp\left(\frac{b}{c+t}\right)$$
(10)$$x^{2}(t)=\frac{2}{\phi_{0}\mu }\int_{0}^{t}\Delta P(t)K(t)dt$$

The above model and similar models have been used with much success by Masoodi et al. [22,75], Languri et al. [76], and Nguyen et al. [77] in predicting the flow in natural fibre reinforcements. This particularly demonstrates the importance of tracking the evolution of the permeability as a function of time in natural fibre reinforcements.

### 3.3. Summary

This section reviewed the developments in the numerical and computational modelling of the mould filling stage in the LCM of plant fibre reinforcements using standard and adapted resin flow models. Traditional physical models such as Darcy’s law and semi-empirical models such as the Carman–Kozeny equation are frequently used to characterise permeability–parameter relationships of plant fibre reinforcements with reasonable accuracy. However, yarn-based plant fibre reinforcements may exhibit substantial influence, including dual-scale flow (inter-yarn macro-flow, and intra-yarn micro-flow) and capillary effects (related to dominant micro-flow); capillary effects in natural fibre reinforcements can be two to three times higher than synthetic fibre reinforcements. The dominance of capillary effects at higher fibre volume fractions (low flow velocities) can lead to inter-yarn voids, while at high flow velocities (and low fibre volume fractions) viscous flow dominates, leading to intra-yarn voids. In addition, deviations from traditional permeability models arise due to the nature of plant fibre reinforcements—they are susceptible to absorbing typical thermosetting resins, such as vinyl esters, and are susceptible to acting as a “sink” and drawing resin away from the main flow. Moreover, the subsequent swelling of plant fibres, restricting flow channels and substantially reducing permeability (by up to 70%), has a “source”-like effects and governs the flow behavior. Most notably, these “sink” and “source” effects, which are unique to plant fibres, are time-dependent. Nonetheless, various adapted flow models have been developed and applied for simulation and verification of the impregnation of full-scale products, such as biocomposite automotive hood parts and chemical storage tanks.

## 4. Conclusions

Permeability studies and data are imperative to understanding and modelling the mould filling stage in the LCM of composites. Plant fibre reinforcements, such as sisal, jute, flax, and hemp fibre mats and woven textiles, demonstrate higher permeability than glass fibre reinforcements, while wood fibre reinforcements exhibit lower permeability than the latter. The permeability of natural fibre reinforcements increases with porosity, and can be modelled by conventional models such as the modified Carman–Kozeny equation. The yarn diameter (linear density), length, and twist level are all found to affect preform permeability, with moderate yarn diamaters, longer fibres, and low-twist reinforcements having higher permeability. Fluid absorption and swelling are key mechanisms in natural fibre preforms, due to which both the saturated and unsaturated permeability are reduced.

For natural preforms, the saturated permeability tends to be higher than the unsaturated permeability, although the difference vanishes with increasing porosity (decreasing fibre volume fraction). Moreover, capillary effects and micro-flow are more dominant in natural fibre reinforcements than in conventional synthetic reinforcements. Importantly, as natural fibres exhibit unique behaviors in comparison to synthetic fibre reinforcements when impregnated by resins (vis. absorption, swelling, dual-scale flow, and capillary effects), in order to accurately measure their permeability, the natural fibre reinforcement should be studied with the particular resin that is to be used during the LCM as the test fluid.

To accurately model the mould filling stage in the LCM of natural fibre reinforcements, researchers have proposed, experimentally validated, modified, and adapted Darcy’s law-based models, which specifically incorporate the effects of liquid absorption (“sink” effect) and subsequent fibre swelling (“source” effect) on preform porosity and permeability as a function of time.

Important insights have been revealed to support better control, optimisation, and simulation of plant fibre composites manufactured by LCM processes and to progress the development of plant fibre composite products in wider applications. Nevertheless, it is evident that much more dedicated research is necessary, particularly at larger length scales for verification of models. Better understanding of the time-dependent characteristics of parameters is also required. There remain a number of gaps in the plant fibre reinforcement permeability and flow behavior literature, including the lack of systematic studies examining time-dependent changes in permeability (due to the sink and source effects), as well as studies using catalysed resins showing viscosity changes as a function of time (and temperature). Most studies in the literature employ oils and glycerin solution as the test fluids, which are incapable of fully representing the flow of a typical liquid thermosetting resin in a plant fibre reinforcement.

## Figures and Tables

**Figure 1 materials-13-04811-f001:**
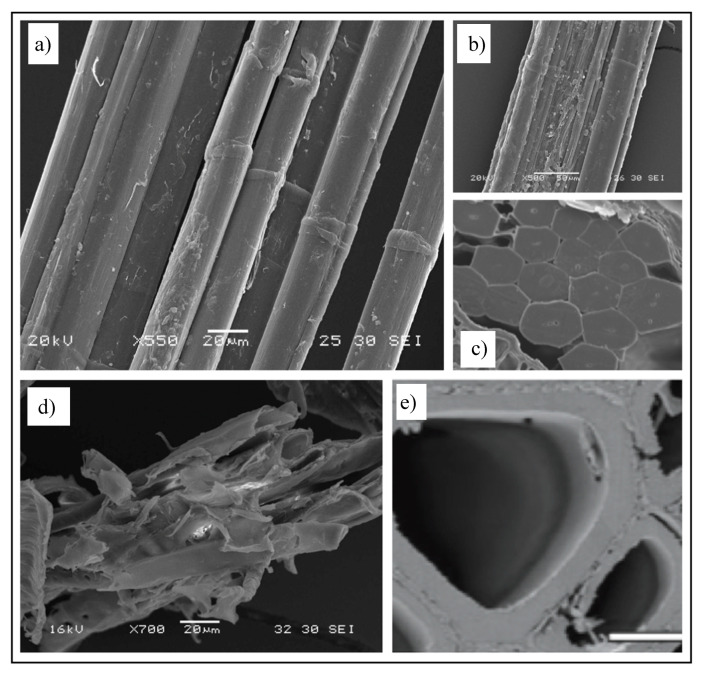
SEM images of plant fibres. High-quality (**a**) and poorly retted (**b**) flax fibres and in planta cross-section (**c**) of flax fibres. (**d**,**e**) Wood fibres. One can notice the difference in lumen sizes between flax and wood. The presence of kink bands in flax is also visible. Images by the author A.B.

**Figure 2 materials-13-04811-f002:**
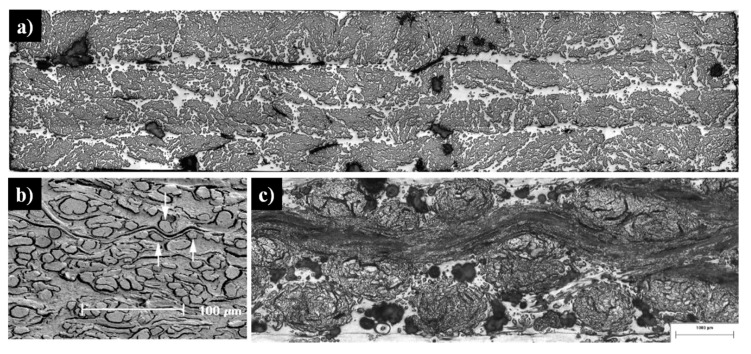
Compaction mechanisms in various plant fibre preforms: (**a**) yarn cross-section deformation, void consolidation, and nesting–packing in a non-woven unidirectional flax composite [14,41,42]. Image by the author DUS. (**b**) Fibre bending and flattening and void consolidation in a random mat wood fibre composite [40]; image reproduced with permission from © 2004 Wiley. (**c**) Yarn cross-section deformation and flattening and nesting–packing in a woven flax composite [43].

**Figure 3 materials-13-04811-f003:**
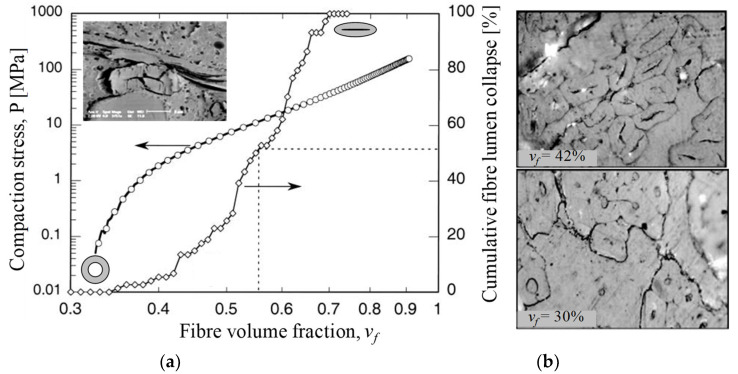
(**a**): Increased compaction pressure on a wood pulp random mat increases the fibre volume fraction, with partial contribution from cumulative lumen collapse-associated fibre deformation [40]; image reproduced with permission from © 2004 Wiley. Inset SEM depicts a collapsed fibre. (**b**): Lumen collapse observed in a woven jute fabric due to increase in compaction pressure at higher fibre volume fractions [13]. Reproduced with permission from © 2011 Sage.

**Figure 4 materials-13-04811-f004:**
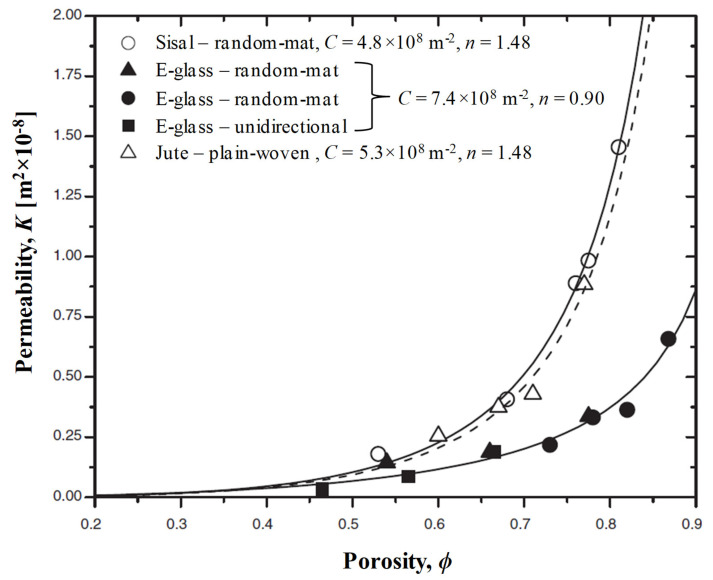
Comparison of the (unsaturated) permeability of sisal, jute, and E-glass reinforcements impregnated with glycerin solution (glycerin/water ratio of 0.88:0.12) at ambient temperature and viscosity of 1.2 Pa∙s. Carman–Kozeny constants (C and n, Equation (5)) are also provided. Adapted from [23], with permission © 2004 Sage.

**Figure 5 materials-13-04811-f005:**
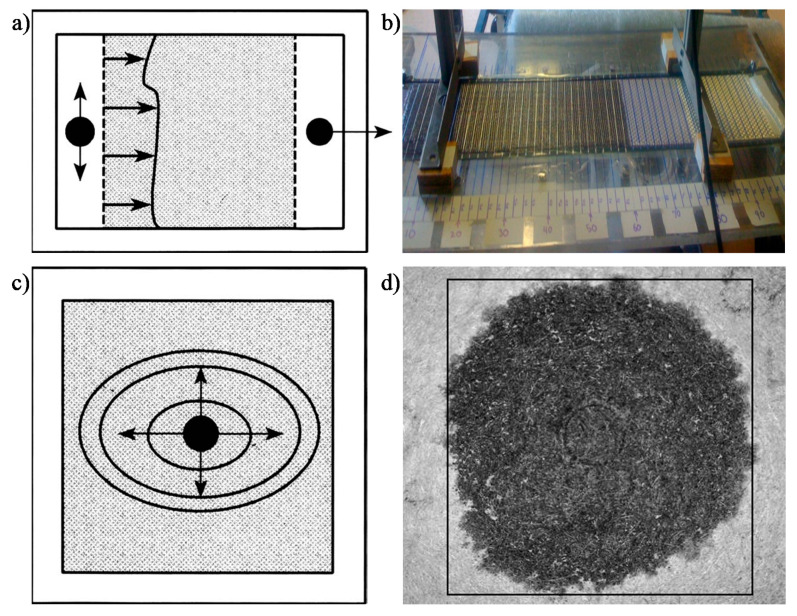
Permeability measurement and flow visualisation setups: (**a**,**b**) 1D linear flow through line gate injection [57]; (**c**,**d**) 2D radial flow through central injection [19,57]; (**a**,**c**) reproduced from [57], with permission from © 1995 Wiley; (**d**) reproduced from [19], with permission from © 2011 Taylor and Francis.

**Figure 6 materials-13-04811-f006:**
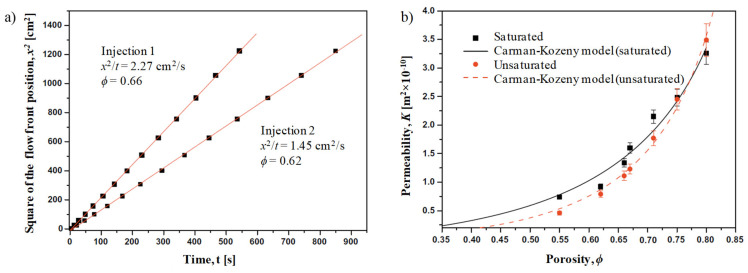
Permeability testing of woven jute fabric with glycerin solution at ament conditions. Adapted from [24], with permission from © 2010, Elsevier. (**a**) Plot of square of flow front position with times for different infusion conditions (related to Darcy’s law). (**b**) Plot of saturated and unsaturated permeability against porosity (= 1 – fibre volume fraction) (related to the Carman–Kozeny equation).

**Figure 7 materials-13-04811-f007:**

Catalysed phenolic resin flow at elevated temperature (60 °C) in 2-layer random mat hemp reinforcements before (**a**) and after (**b**) fibre washing and edge flow problems were resolved. The flow front is smoother, more uniform, and quasi-1D in (**b**). Flow front isochrones are shown, where each isochrone represents 20 s. The mould filling direction is from left to right. Reproduced from [25], with permission from © 2000 Elsevier.

**Figure 8 materials-13-04811-f008:**
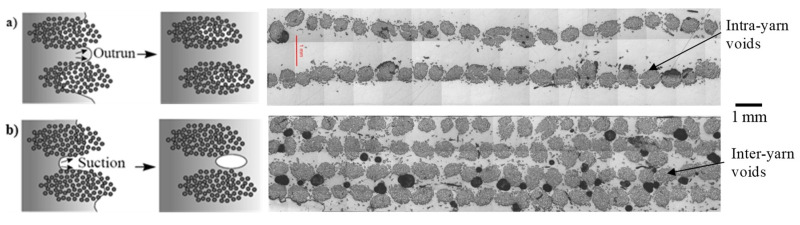
The complex dual-scale flow of resin in fibrous preforms can generate voids. Macro-scale flow relates to the advance of resin between yarns or tows (i.e., inter-yarn flow), while micro-scale flow relates to the penetration of resin into a yarn (i.e., intra-yarn flow). Note that permeability is also different at the two scales. For instance, (**a**) for low fibre content, due to low yarn permeability but high overall permeability, the yarn is not properly impregnated, and thus intra-yarn voids may form, while (**b**) for high fibre content, although the yarn and overall permeability are similar, capillary flow in the yarn dominates, and therefore inter-yarn voids are formed. Images reproduced by author D.U.S. from [41,63], showing jute and flax unidirectional fabrics impregnated by epoxy and polyester resins at ambient conditions.

**Figure 9 materials-13-04811-f009:**
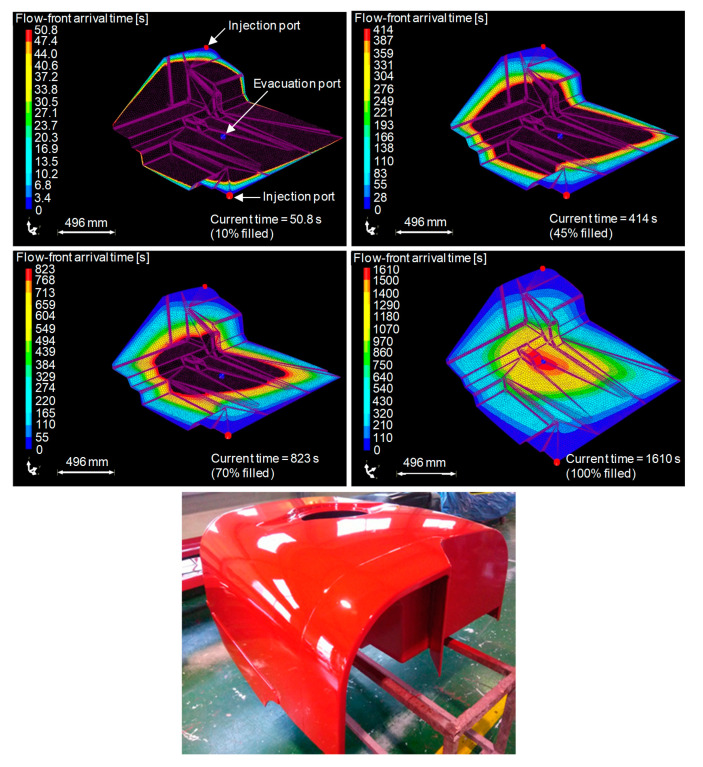
Resin flow simulation conducted in RTM-Worx software for the vacuum-assisted, light-RTM manufacture of the lower part of a flax–vinylester composite agricultural chemical storage tank. Images show progression of the flow front at time intervals as the part is 10%, 45%, 70% and 100% filled, with an image of the final fabricated component. Adapted from [64], with permission from © 2014 Prof. C Kong.

**Table 1 materials-13-04811-t001:** The dominant and secondary compaction mechanisms at the fibre, yarn, and fabric scales in random mat, non-woven, and woven preforms.

Scale	Mechanism		Dominant in	Secondary in
Fibre/filament 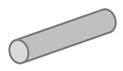	cell wall/lumen collapse (fibre cross-section deformation)	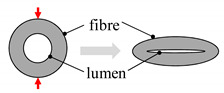	all plant fibre preforms	-
Yarn/tow, Unidirectional tape 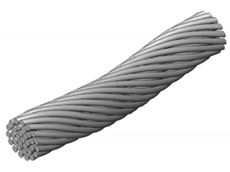	yarn cross-section deformation	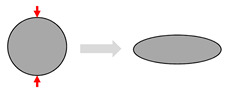	all	-
	void condensation (i.e., closing gaps between fibres)	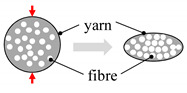	random mat, unidirectional tape	woven and non-woven
	yarn flattening	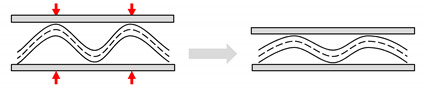	woven	random mat
	fibre/yarn bending deformation	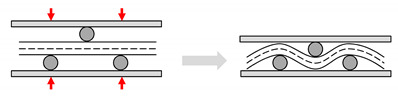	-	all
Fabric/preform 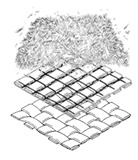	nesting and packing	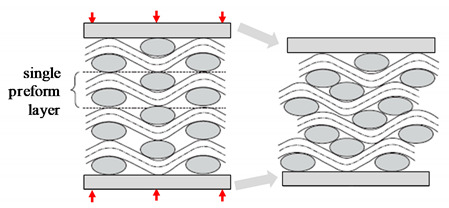	woven	non-woven

**Table 2 materials-13-04811-t002:** In-plane permeability data for various plant fibre reinforcements. Carman–Kozeny constants (C and n, Equation (5)) are provided. It is specified whether data are for transverse or through-thickness saturation.

Reinforcement	*C* [×10^8^ m^−2^]	*n*	*K* at *v_f_* = 0.2 (or *ϕ* = 0.8) [×10^−8^ m^−2^]	*K* at *v_f_* = 0.5 (or *ϕ* = 0.5) [×10^−8^ m^−2^]	Unsaturated or Saturated, Test fluid, Viscosity and Temperature	Source
**Wood fibre—random mat**	2460	1.80	0.00394	0.000203	Saturated Mineral oil, 0.066–0.095 Pa·s, 14–23 °C	[46]
**Wood fibre—random mat**	4000	1.76	0.00229	0.000125	Saturated Mineral oil, 0.066–0.095 Pa·s, 14–23 °C	[46]
**Sisal—** **random mat**	4.8	1.48	1.30	0.104	Unsaturated, Glycerin solution, 1.2 Pa·s, ambient	[23]
**Jute—** **plain-woven**	5.3	1.48	1.17	0.0943	Unsaturated, Glycerin solution, 1.2 Pa·s, ambient	[23]
**Sisal—** **plain-woven**	22.5	2.00	0.569	0.0222	Unsaturated, Vinylester resin, 0.5–0.9 Pa·s	[47]
**Jute—** **plain-woven**	81.0	0.88	0.0335	0.00617	Saturated, Glycerin solution, 0.13 Pa·s, ambient	[17]
**Jute—** **plain woven**	133.8	1.29	0.0357	0.00373	Unsaturated, Glycerin solution, 0.15 Pa·s, ambient	[24]
**Jute—** **plain woven**	84.6	0.91	0.0334	0.00591	Saturated, Glycerin solution, 0.15 Pa·s, ambient	[24]
**Coconut—** **random mat**	0.21	1.45	0.0160	-	Unsaturated, Glycerin solution, 0.085 Pa.s, ambient	[48]
**Jute—** **plain-woven**	-	-	0.00351	0.00197	Saturated Mineral oil, 0.180 Pa.s, ambient	[49]
**Jute—** **plain-woven**	-	-	0.00270	0.00060	Saturated, corn syrup solution, 0.180 Pa.s, ambient	[49]
**Flax–** **random mat**	-	-	0.00028	0.00005	*Transverse* saturated, Silicon oil, 0.1 Pa.s, ambient	[50]
**Flax–** **plain-woven**	-	-	0.00090 (at *v_f_* = 0.3)	0.00006	*Transverse* saturated, mineral oil, 0.14 Pa.s, 20 °C	[51]
**Flax–** **plain-woven**	-	-	0.080 (at *v_f_* = 0.3)	0.00030	K1 In-plane saturated, mineral oil, 0.14 Pa.s, 20 °C	[51]

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
