# Peer review of "A Review of Permeability and Flow Simulation for Liquid Composite Moulding of Plant Fibre Composites"

_materials, 2020, doi:10.3390/ma13214811_

Round 1

Reviewer 1 Report

The review manuscript describes the processing parameters for LCM of plant fiber composites. It is well-organized with plenty of information. However, there are some to further improve the manuscript below.

  1. The authors had better deal further with curing conditions after impregnation to make this manuscript be involved with the whole processing section although it is not necessary.
  2. The figures and referenced works should contain the information in detail on matrix materials, temperature, etc. These pieces of information are crucial for the understanding of each composite system.
  3. How is the effect of catalyst along with temperature on permeability and flow? The catalyst at high temperatures may influence them.
  4. The authors focused on porosity and fiber volume fractions. How are the effects of the fiber diameter/length and their correlation?
  5. Can you compare the effects of the resin viscosities at high and low temperatures?

Author Response

Reviewer 1 and 3 have proposed acceptance following minor revisions, and Reviewer 2 has recommended publication in its present form. Revisions in accordance to the suggestions of Reviewer 1 and 3 have been made to the manuscript. A point-by-point response to the referee’s comments is presented in this document; when referring to text in the manuscript, we use an italicised font. A tracked changes version of the manuscript has been submitted, alongside a version with the changes all-accepted. Reviewer 4 recommend the article not to published, and did not propose modifications and areas of improvement. Particularly as Reviewer 4 has not offered any coherent modifications to the manuscript, we on principle, will not be responding to them. Reviewers 1,2 and 3 are overwhelming positive and demonstrate the significance and coherence of our manuscript.

Reviewer 1 – minor revisions

Comments:

The review manuscript describes the processing parameters for LCM of plant fiber composites. It is well-organized with plenty of information. However, there are some to further improve the manuscript below.

Response:

Many thanks for your detailed, encouraging and positive review of our work. We have made revisions to our manuscript based on your suggestions.

Comments:

The authors had better deal further with curing conditions after impregnation to make this manuscript be involved with the whole processing section although it is not necessary.

The figures and referenced works should contain the information in detail on matrix materials, temperature, etc. These pieces of information are crucial for the understanding of each composite system.

How is the effect of catalyst along with temperature on permeability and flow? The catalyst at high temperatures may influence them.

Can you compare the effects of the resin viscosities at high and low temperatures?

Response:

The Reviewer has asked to incorporate more information on details of matrix materials, temperatures, viscosities, as well as curing conditions of the referenced works.

In Table 2, for example, the test fluid (glycerine solution, oil or specific resin such as vinylester) and the tested viscosity is clearly presented.       

Most permeability studies on plant fibre composites have followed methodologies adopted by permeability on synthetic fibre composites. What was not previously clear and now has been explained in the text early in Section 2.3 is that: ‘Predominant number of studies have used mineral oil (non-polar fluid) or non-reactive glycerine solution (polar fluid), at fixed (often ambient) temperatures and viscosities (comparable to commercial resins), to study the permeability of plant fibre reinforcements. While the use of oils enable the study of permeability with no fluid absorption and consequent swelling of the plant fibres, the use of glycerine enables the study of the sink and source effects (described in more detail in Section 3.2). A few studies have examined permeability and/or flow behaviour with commercial thermosetting resins such as phenolics [22] at elevated temperatures (e.g. 60 °C). Though even in these cases viscosity is (assumed to be) constant during the mould filling stage as fill-time (of 60 s to 600 s) is significantly shorter than the extended pot life (several hours) enabled by selected catalysts.’ Hence, earlier, we had not presented the requested information, as most of it is not yet studied or available. Even temperature is often not reported (and is assumed to be ambient).

We have updated the manuscript, including Table 2 and figure captions, to include detail on temperatures, fluids, viscositie etc. However, as discussed, there is not much (if any) research on the effects of resin viscosities at high and low temperatures, and then influence of catalysts etc. We have now extended the conclusions, and included a paragraph on gaps in literature – data on the effects of resin viscosities at high and low temperatures is a gap.

Of course, the effect of fluid viscosity on permeability and flow behaviour is particularly apparent in Section 3 on modelling, as viscosity is a key parameter in the governing equations.

Comments:

The authors focused on porosity and fiber volume fractions. How are the effects of the fiber diameter/length and their correlation?

Response:

We agree. Hence, the conclusions of our review paper clearly state that: ‘Yarn diameter (linear density), length and twist level are all found to affect preform permeability.’ We note here that while there is research on the effects of yarn diameter and fibre length on permeability, there is no research on the effect of plant fibre diameter (elementary fibres or fibre bundles) on permeability – indeed, given the natural variability in fibre diameter, this is much harder to control, than say length (which can be easily cut).

As a detailed discussion, Section 2.3 includes the effects of fibre length:

…The argument may have merit as flax yarn random-mats composing of longer fibres have notably and consistently higher (saturated) permeability across a range of fibre volume fractions [16]. For instance, the permeability of random-mats with 50 mm fibre lengths was found to be 22% and 25% higher at the lowest (vf = 0.2) and highest (vf = 0.4) fibre volume fractions, in comparison to random mats with 15 mm fibre lengths [16]. Shives, present in flax mats, may also increase the permeability, due to their porous structure, by creating channels for fluid movement, blurring the influence of the volume fraction [47].

            Section 2.3 also includes detail on the effects of yarn diameter:

Yarn diameter (or linear density) is also known to affect the permeability of flax yarn random-mats [16].  Umer et al. [16] found that the permeability of medium yarn diameter (0.56 mm) mats was consistently (i.e. over a range of porosity levels) 27% higher than small yarn diameter (0.35 mm) mats, but the permeability of small and medium yarn diameter mats was consistently 68-77% higher than large yarn diameter (0.81 mm) mats. Given that the permeability of preforms is dominated by the characteristics of open channels, and that preform geometric parameters that describe the characteristics of the open channels include number of fibres in the bundle, twist angle and orientation of bundle, dimension and cross-section shape of bundles and single fibres, Umer et al. suggested that large diameter yarns were less compact and had lower twist levels. The loose fibres on the surface of large diameter yarns restrict fluid flow through the open channels, thereby decreasing permeability. The twist level of yarns may also affect permeability and impregnability by altering competition between micro-flow and macro-flow [16,36,48]. This dual-scale flow is reviewed further in a later section.

Similarly, Section 3.2.3 on modelling also says:

‘… Masoodi et al. [18] proposed that the permeability K, alongside porosity Ï• and fibre diameter d, were only functions of time (Eq. 9), where the subscript 0 indicates initial values (at t = 0). Essentially, if the change in fibre diameter over time (due to swelling) was measured, fitted curves could be used to then predict evolution of porosity with time, and the evolution of permeability with time.’

Reviewer 2 Report

A preliminary reading of the review titled “A review of permeability and flow simulation for liquid composite moulding of plant fibre composites” offers a very good impression to the reader. Even more, this positive impression is confirmed by far after a more profound reading of the article. This reviewer has enjoyed very much it.

The introduction section is very well organized, paying attention to some aspects usually not considered in other reviews based on vegetal fiber as reinforcement. The fundamentals of LCM processes are welcomed since they are not trivial for many potentials readers of Materials. Good job!!

The key points played by the plant fibers in LCM are nicely pointed out. The particularities of these, their possible morphologies, lumen dimensions, and so on, jointly to the possible fiber surface properties become a superb task by the authors. Brave!!

The concerning to the LCM process is also well explained. The authors consider plenty of fiber reinforcements paying also attention to the permeability and anisotropy in the ultimate properties. Well done again!!

The third section is related to the flow modeling and simulations, paying attention to classical flow models, the duel-scale flow and capillarity, and to the modification of the flow models to realistic considers the fiber particularities. Nice work!!

The conclusions are concise and well-founded.

The reference section seems adequate and up-to-date, including a 15% of very recent citations (of the two last years).

With the above comments, this reviewer is glad to recommend the PUBLICATION OF THE ARTICLE in its actual state.

Author Response

Reviewer 1 and 3 have proposed acceptance following minor revisions, and Reviewer 2 has recommended publication in its present form. Revisions in accordance to the suggestions of Reviewer 1 and 3 have been made to the manuscript. A point-by-point response to the referee’s comments is presented in this document; when referring to text in the manuscript, we use an italicised font. A tracked changes version of the manuscript has been submitted, alongside a version with the changes all-accepted.

Reviewer 2 – publish in present state

Comments:

A preliminary reading of the review titled “A review of permeability and flow simulation for liquid composite moulding of plant fibre composites” offers a very good impression to the reader. Even more, this positive impression is confirmed by far after a more profound reading of the article. This reviewer has enjoyed very much it.

The introduction section is very well organized, paying attention to some aspects usually not considered in other reviews based on vegetal fiber as reinforcement. The fundamentals of LCM processes are welcomed since they are not trivial for many potentials readers of Materials. Good job!!

The key points played by the plant fibers in LCM are nicely pointed out. The particularities of these, their possible morphologies, lumen dimensions, and so on, jointly to the possible fiber surface properties become a superb task by the authors. Brave!!

The concerning to the LCM process is also well explained. The authors consider plenty of fiber reinforcements paying also attention to the permeability and anisotropy in the ultimate properties. Well done again!!

The third section is related to the flow modeling and simulations, paying attention to classical flow models, the duel-scale flow and capillarity, and to the modification of the flow models to realistic considers the fiber particularities. Nice work!!

The conclusions are concise and well-founded.

The reference section seems adequate and up-to-date, including a 15% of very recent citations (of the two last years).

With the above comments, this reviewer is glad to recommend the PUBLICATION OF THE ARTICLE in its actual state.

Response:

Many thanks for your detailed, encouraging and positive review of our work. Many thanks also for recommending publication of our article in its present state.

Reviewer 3 Report

Topic of the article "A review of permeability and flow simulation for
3 liquid composite molding of plant fiber composites is interesting and may attract readers' interest. Nevertheless, the text needs to be improved. I recommend minor revision:

  • The main aim of the article should be indicated in the introduction or abstract.
  • When defining LCM composite, it should be first specified which polymers are used in this technology. Please provide some examples.
  • The introduction should be supplemented with literature items describing fiber-reinforced composites from 2019-2020. For example:

A review on plant fiber reinforced thermoset polymers for structural and frictional composites, https://doi.org/10.1016/j.polymertesting.2020.106792

Hybrid Epoxy Composites with Both Powder and Fiber Filler: A Review of Mechanical and Thermomechanical Properties, https://doi.org/10.3390/ma13081802.

Flax fiber and its composites: An overview of water and moisture absorption impact on their performance, https://doi.org/10.1177/0731684418818893

Assessment of capillary phenomena in liquid composite molding,https://doi.org/10.1016/j.compositesa.2019.02.018

  • The authors did not refer to Figure 1A in the text.
  • The phrase "textile reinforcements" is unclear and should be clarified.
  • Figure 4 should be increased in size to make it easier to read.
  • It should be in the description of point 2.3 explain what the symbol C means in table 2, because the explanation appears only with equation 5.
  • Each of the main points, ie 1 and 2 and 3, should be ended with a summary paragraph, which would make it easier for the reader to use the collected information.
  • The conclusion should include specific examples of composites (type of reinforcement / polymer used) that exhibit specific permeability tends.
  • The summary of the review article should contain more practical information that the reader will be able to use in the process of producing LCM materials

Author Response

Reviewer 1 and 3 have proposed acceptance following minor revisions, and Reviewer 2 has recommended publication in its present form. Revisions in accordance to the suggestions of Reviewer 1 and 3 have been made to the manuscript. A point-by-point response to the referee’s comments is presented in this document; when referring to text in the manuscript, we use an italicised font. A tracked changes version of the manuscript has been submitted, alongside a version with the changes all-accepted. Reviewer 4 recommend the article not to published, and did not propose modifications and areas of improvement. Particularly as Reviewer 4 has not offered any coherent modifications to the manuscript, we on principle, will not be responding to them. Reviewers 1,2 and 3 are overwhelming positive and demonstrate the significance and coherence of our manuscript.

Reviewer 3 – minor revisions

Comments:

Topic of the article "A review of permeability and flow simulation for liquid composite molding of plant fiber composites is interesting and may attract readers' interest. Nevertheless, the text needs to be improved. I recommend minor revision.

Response:

Many thanks for your detailed, encouraging and positive review of our work. We have made revisions to our manuscript based on your suggestions.

Comments:

The main aim of the article should be indicated in the introduction or abstract.

Responses:

Agreed - Thank you for this useful suggestion. We have now included ‘we aim to..’ in the abstract, and in more detail express the main aim of our review in the final paragraph of the introduction (line 90-94):

An overarching aim of our review is to support better control, optimization and simulation of plant fibre composites manufactured by LCM processes, and thereby progress the development of plant fibre composite products in wider applications with reduced waste and efficient processing. The review has been written for an informed composites audience, and will be particularly of interest to the biocomposites community, in academia and industry.

Comments:

When defining LCM composite, it should be first specified which polymers are used in this technology. Please provide some examples.

Responses:

Agreed. The second paragraph (line 35-36) of the introduction now lists examples of thermosets used in the technology:

The basic approach in any LCM process is to force a catalysed thermosetting liquid resin, such as epoxy, polyester, vinylester, phenolic, and furan resin, to…

Comments:

The introduction should be supplemented with literature items describing fiber-reinforced composites from 2019-2020. For example:

  • A review on plant fiber reinforced thermoset polymers for structural and frictional composites, https://doi.org/10.1016/j.polymertesting.2020.106792
  • Hybrid Epoxy Composites with Both Powder and Fiber Filler: A Review of Mechanical and Thermomechanical Properties, https://doi.org/10.3390/ma13081802.
  • Flax fiber and its composites: An overview of water and moisture absorption impact on their performance, https://doi.org/10.1177/0731684418818893
  • Assessment of capillary phenomena in liquid composite molding, https://doi.org/10.1016/j.compositesa.2019.02.018

Responses:

Agreed and many thanks for some of these references – we have incorporated them. These have been included in relevant sections of the introduction, on resin absorption, and capillarity effects. Reference two, however, on ‘hybrid epoxy composites with both powder and fiber filler’ was not entirely relevant to any section of our review, so we have not included that specific reference.

Comments:

  • The authors did not refer to Figure 1A in the text.
  • The phrase "textile reinforcements" is unclear and should be clarified.
  • Figure 4 should be increased in size to make it easier to read.
  • It should be in the description of point 2.3 explain what the symbol C means in table 2, because the explanation appears only with equation 5.
  • Each of the main points, ie 1 and 2 and 3, should be ended with a summary paragraph, which would make it easier for the reader to use the collected information.

Responses:

Agreed - we have made these additional minor revisions.

  • Figure 1A and 1B are now first referred to in the text: ‘The surface roughness of the fibre may be affected by the fibre extraction methods, and particularly the quality of the retting process (Figure 1A,B).
  • The text in lines 62-64 clarifies what textile reinforcement mean:
    …utilising textile reinforcements, which are fabrics comprising of aligned, continuous yarns/tows that are knitted/, woven/, stitched/, or braided, in thermosetting matrices at high fibre content.
  • As recommended, Figure 4 has now been increased in size.
  • Agreed and apologies for missing this out earlier – we have now added a brief description of C and n in Section 2.3.
  • Thank you for this suggestion – we have now included a Section 2.4 and Section 3.3 with a short summary at the end of each major sections. Section 1 has a paragraph ending with the main aims and structure of the review.

Comments:

  • The conclusion should include specific examples of composites (type of reinforcement / polymer used) that exhibit specific permeability tends.
  • The summary of the review article should contain more practical information that the reader will be able to use in the process of producing LCM materials

Responses:

      We have now included short sections, Section 2.4 and Section 3.3, to summarise the major sections. These offer more specific examples. We do not want to replicate too much in summary and conclusion sections, as the paper is already very lengthy. We have extended the conclusion to now also include what we think of as gaps in literature.

Reviewer 4 Report

This work is supposed to be a review of LCM process ass applied to natural fiber composites. The LCM process as a means of fabricating large composite components has been in use for over 30 years, and the science underpinning the technology is well advanced. I do not see anything new offered by the present study, besides repeating results well known in the field, form a very narrow point of view. The authors are clearly out of their depth in terms of flow through fibrous media - key contributing references are absent  eg. Advani and co-workers, Gebart and co-workers, Papathanasiou et al., Matsumura et al., Lundstrom and co-workers, Yazdchi et al. etc etc - and the reference to the dual-scale issue is simply a simplistic representation of  material that has been well reviewed and analyzed in the 90s. As an additional example, extensive reference is made to the determination of the Kozeny constant, while it is known that the Carman-Kozeny equation is NOT a suitable model for the permeability of fibrous media, for which the measured Cozeny constant varies erratically. I recommend the authors study this matter further, build on this preliminary literature survey and produce their own results, instead of attempting to present it as an authoritative Review article. The recommendation is to not publish.

Author Response

Reviewer 1 and 3 have proposed acceptance following minor revisions, and Reviewer 2 has recommended publication in its present form. Revisions in accordance to the suggestions of Reviewer 1 and 3 have been made to the manuscript. A point-by-point response to the referee’s comments is presented in this document; when referring to text in the manuscript, we use an italicised font. A tracked changes version of the manuscript has been submitted, alongside a version with the changes all-accepted. Reviewer 4 recommend the article not to published.

Reviewer 4's Comments:

This work is supposed to be a review of LCM process ass applied to natural fiber composites. The LCM process as a means of fabricating large composite components has been in use for over 30 years, and the science underpinning the technology is well advanced. I do not see anything new offered by the present study, besides repeating results well known in the field, form a very narrow point of view. The authors are clearly out of their depth in terms of flow through fibrous media - key contributing references are absent  eg. Advani and co-workers, Gebart and co-workers, Papathanasiou et al., Matsumura et al., Lundstrom and co-workers, Yazdchi et al. etc etc - and the reference to the dual-scale issue is simply a simplistic representation of  material that has been well reviewed and analyzed in the 90s. As an additional example, extensive reference is made to the determination of the Kozeny constant, while it is known that the Carman-Kozeny equation is NOT a suitable model for the permeability of fibrous media, for which the measured Cozeny constant varies erratically.

I recommend the authors study this matter further, build on this preliminary literature survey and produce their own results, instead of attempting to present it as an authoritative Review article. The recommendation is to not publish.

Response:

We are surprised that the Reviewer makes statements attacking the authors personally such as ‘the authors are clearly out of their depth…’ – this is beyond the remit of a review.

Moreover, the Reviewer does not make any propositions to improve/revise our submitted review manuscript - other than 'I recommend that authors...produce their own results...' - hence there isn’t anything specific we can respond to.

We have tried to articulate the aims and objectives of the article better. We would like to point out to the reviewer that this is NOT a review of LCM processes (for which the works of Advani, Gebart etc would be very relevant). This is a review of LCM SPECIFIC TO plant fibre reinforcements, which, as we demonstrate, are a totally different beast in comparison to traditional synthetic fibres. Indeed, we do include references to the work of Advani (such as Reference [65] with Pillai and Advani), and others, who have done work relevant to plant fibre composites.

The Reviews of Reviewer 1, 2 and 3 are overwhelmingly positive and supportive of our work – they demonstrate the significance, relevance and impact of our work.

Round 2

Reviewer 1 Report

The revised manuscript followed the reviewer's comments and the manuscript has been quite improved.

Reviewer 4 Report

The authors have not responded to my comments, mainly related to what I perceive to be an incomplete coverage of the field of LCM, citing instead the positive reviews received by other reviewers. I therefore stand by my original review. The authors claim that natural fibers are a different class of materials, compared to synthetic fibers. This is true in some respects (swelling, deformability, eg Fig. 4). However, there is a great deal of similarities as well, and the manuscript is largely based on ideas/concepts that have originated and are being used in the filed of LCM of synthetic fibers. I think that a review article, to serve the community, must have a true global understanding of the related fields (LCM and fibrous materials here) to properly give perspective and assist young researchers. I dont believe the manuscript in consideration helps in this direction.